# Surface Modification of Polyetheretherketone (PEEK) Intervertebral Fusion Implant Using Polydopamine Coating for Improved Bioactivity

**DOI:** 10.3390/bioengineering11040343

**Published:** 2024-03-31

**Authors:** Suzy Park, Tae-Gon Jung

**Affiliations:** Medical Device Development Center, Osong Medical Innovation Foundation, 123 Osongsaengmyung-ro, Osong-eub, Heungdeok-gu, Cheongju-si 28160, Chungbuk, Republic of Korea; owner5306@kbiohealth.kr

**Keywords:** biomaterial, bone tissue engineering, polydopamine, PEEK, surface modification

## Abstract

The occurrence of bone diseases has been increasing rapidly, in line with the aging population. A representative spinal fusion material, polyetheretherketone (PEEK), is advantageous in this regard as it can work in close proximity to the elastic modulus of cancellous bone. However, if it is used without surface modification, the initial osseointegration will be low due to lack of bioactivity, resulting in limitations in surgical treatment. In this study, we aimed to modify the surface of PEEK cages to a hydrophilic surface by coating with polyethylene glycol (PEG), hyaluronic acid (HA), and polydopamine (PDA), and to analyze whether the coated surface exhibits improved bioactivity and changes in mechanical properties for orthopedic applications. Material properties of coated samples were characterized and compared with various PEEK groups, including PEEK, PEEK-PEG, PEEK-HA, and PEEK-PDA. In an in vitro study, cell proliferation was found to be enhanced on PDA-coated PEEK; it was approximately twice as high compared to the control group. In addition, mechanical properties, including static and torsion, were not affected by the presence of the coating. Thus, the results suggest that PEEK-PDA may have the potential for clinical application in fusion surgery for spinal diseases, as it may improve the rate of osseointegration.

## 1. Introduction

In recent years, a rapidly aging population has led to an increase in the occurrence of bone diseases, including bone cancer and bone defects due to accidents. Within this field, intervertebral fusion prosthesis is widely used to treat spinal diseases such as intervertebral stenosis, intervertebral disc prolapse, and posterior joint hypertrophy. Such spinal conditions are increasing due to an aging society, and various types of prostheses are being developed, differing in their design, procedure method, and material [1].

Among them, the intervertebral fusion cage is a medical device used in spinal fusion; it is inserted between the joints of the spine to restore the gap between the vertebral bodies and to secure space for surgery. Depending on the material used, there will be various differences in mechanical properties such as strength, elasticity, and ductility, as well as in biological properties such as bio-friendliness and bone union [2,3]. 

Implant development using a polymer similar to human bone has recently become available, instead of using a metallic material with a high elastic modulus (50 GPa). For instance, polymer materials such as PEEK, which has an elastic modulus of 4 GPa and transmits radiation, have begun to attract attention as high-performance biomaterials to replace metallic materials [4,5,6]. PEEK does not dissolve in other organic solvents (excluding sulfuric acid) due to its stable chemical structure. It also exhibits excellent mechanical properties, heat resistance, corrosion resistance, and biocompatibility [7,8].

To overcome this problem, PEEK has been fabricated using various surface modifications such as coating methods [9,10]. We aimed to provide a bioactive function by using a coating technique on PEEK with various biopolymers. In particular, we assumed that it would improve the initial rate of osseointegration onto PEEK using polyethylene glycol (PEG), hyaluronic acid (HA), and polydopamine (PDA) polymers. 

PEG is a hydrophilic polymer that is soluble in both organic solvents and water, exhibits non-toxicity and non-immunity properties in the body, and has low interfacial free energy when in contact with water. These properties play a role in preventing the non-specific adsorption of biomaterials such as proteins and cells on various surfaces. Therefore, it is widely used as an excellent biocompatible material for the surface modification of polymer materials such as PEEK implants [11,12]. 

Also, HA has excellent biocompatibility and biodegradability because it is a fundamental component of the living body [13,14]. It can be developed into more functionally improved materials through immobilization on medical implants including dental implants, stents, screws, and bone substitutes, or on the surface of medical devices [15,16]. 

PDA can be coated on super-hydrophobic surfaces; it is known to adhere strongly to the surfaces of many metals, nonmetals, organic polymers, and inorganic materials, including polytetrafluoroethylene (PTFE) [17,18,19]. In particular, catechol, a chemical functional group of PDA, derives oxidation and reduction capabilities on the surface. It can react chemically with the amine functional group to form a covalent bond on the surface modified with the amine functional group, enabling secondary surface modification. Catechol is a natural, non-toxic material actively used in cell and tissue engineering because of its biodegradability [20,21,22]. 

Herein, we discuss various analysis experiments undertaken to select the most bioactive condition among the polymers used to form a coating layer for clinical application. In brief, various PEEK groups were fabricated with biopolymers (i.e., PEG, HA, and PDA), as depicted in Figure 1. The biocompatibility and physicochemical and mechanical characteristics of various PEEK groups were observed and compared. 

## 2. Materials and Methods

### 2.1. Preparation of PEEK 

The PEEK (VESTAKEEP i4 R, EVONIK, Essen, Germany) used in this study is a medical PEEK material and was prepared in two forms: as a disk with a diameter of 10 mm and a thickness of 5 mm for surface characteristic analysis and the cell activity assay, and as an intervertebral fusion prosthesis for mechanical tests. Before the experiment, to maintain stable coating quality and confirm that the surface condition was optimized for coating, ultrasonic washing was performed with 12.5% acetone solution, and hot air drying was performed at 80 °C for 30 min for the specimen pretreatment process.

### 2.2. Preparation of the PEEK-Modified Surface

The coating materials used for the surface modification of PEEK included PEG (Polyethylene glycol 400, Junsei Chemical Co., Ltd., Tokyo, Japan), HA (Sodium Hyaluronate, Bloomage Freda Biopharm Co., Ltd., Jinan, China), and PDA (Polydopamine Hydrochloride, Sigma-Aldrich Pte. Ltd., St. Louis, MO, USA). A solution was prepared with a concentration of 2 wt% each of PEG and HA, while PDA was used at a concentration of 2 mg/mL (Tris-buffer, pH 8.0). And then, the PEEK samples were immersed in various solutions (PEG, HA, PDA) for 24 h. The above process was carried out in the same way for both PEEK in the form of an intervertebral fusion prosthesis and PEEK in the form of a disk.

### 2.3. Characterizations

#### 2.3.1. Physicochemical Characterization

To validate the uniform coating of PEG, HA, and PDA on the PEEK surface, the PEEK surface was observed using a field emission scanning electron microscope (FE-SEM, TESCAN MIRA3, Kohoutovice, Czech Republic). The surface shape and microstructure of PEEK before and after coating were confirmed and analyzed at magnifications of ×500 and ×5000.

In addition, to confirm the coating layers of PEG, HA, and PDA on the PEEK surface, they were investigated using the focused ion beam technique (FIB, ThermoFisher, Helios 5 UC, Waltham, MA, USA). To avoid damage to the various PEEK groups during the FIB milling process, the surface of the PEEK groups was sputtered with a thin layer of platinum before the milling process. 

Fourier transform infrared spectroscopy (FT-IR) was carried out using an infrared spectrophotometer (FT-IR spectrophotometer, Thermo Scientific^TM^, Waltham, MA, USA) and the results were compared before and after coating. FT-IR measurements were undertaken in the wavelength range of 500 cm^−1^ to 4000 cm^−1^, and the frequency region of 600 cm^−1^ to 1200 cm^−1^ and the functional group frequency region of 1200 cm^−1^ to 2000 cm^−1^ were analyzed.

The surface average roughness (R_a_), average square roughness (R_q_), and 10-point average roughness (R_z_) values were measured using a surface roughness measurement system (Mitutoyo SJ-410, Kawasaki, Japan) to analyze the surface properties of PEEK. A comparison was made between the PEEK surface before and after coating. Measurement conditions were measured following ISO 1997 standards.

The contact angle of each sample was measured using the water drop method with equipment (CA, SEO Phoenix 300, Seoul, Republic of Korea). The contact angle was measured by dropping 5 µL of distilled water droplets on each coating surface with PEEK. The arithmetic mean contact angle was calculated and compared through repeated measurements, and conducted more than five times depending on the surface. 

#### 2.3.2. In Vitro Cell Culture and Biocompatibility

To measure the proliferation of osteoblast cells (MC3T3–E1) on the surface of the PEEK disk, the cells were sub-cultured three times using a DMEM (Dulbecco’s Modified Eagle’s Medium) culture containing 10% fetal bovine serum and 1% penicillin-streptomycin. The cells were dispensed onto PEEK disks under each condition at a concentration of 1.0 × 10^6^, and subsequently incubated for seven days in a 5% CO_2_ environment using DMEM culture at 37 °C in an incubator. 

The biocompatibility of each sample was measured using a WST-1 cell cytotoxic assay kit (Takara Bio, Otsu, Japan). 

In summary, 

(1)WST-1 solution (30 µL) and the free DMEM (300 µL) were mixed and then added to each sample.(2)They underwent additional incubation at 37 °C for 30 min.(3)The solutions of each sample were carried to a 96-well plate and the absorbance at 450 nm was observed by using a microplate reader (Bio-Rad Laboratories, Hercules, CA, USA).(4)The percentage of cell viability was calculated as follows and the five specimens were subsequently analyzed repeatedly for each condition at three different times.


(1)
$$Cell\;viability\left(\%\right)=\left[\frac{\left(A_{S}\right)-\left(A_{b}\right)}{\left(A_{C}\right)-\left(A_{b}\right)}\right]\times 100$$


*As* = Absorbance (absorbance of cells);

*Ab* = Absorbance of the blank (absorbance of medium and WST-1);

*Ac* = Absorbance of the control (absorbance of containing cells, medium, and WST-1).

#### 2.3.3. Mechanical Tests 

For the mechanical test, axial compression and axial torsion tests were performed using a universal testing machine (MTS E45, MTS Systems, MN 55344-2247, Eden Prairie, MN, USA). For each test, six specimens were prepared for each coating. In addition, a jig suitable for the specimen of the intervertebral fusion prosthesis PEEK was manufactured. The test was performed using the ASTM F 2077 (Test Methods for Intervertebral Body Fusion Devices for axial compression/torsion test) [23].

For the static axial compression test, the experiments were performed according to the ASTM F 2077. Measurement conditions were measured by fixing the specimen to the jig and applying a load at a rate of 25 mm/min until functional or until the mechanical failure of the intervertebral fusion implant through the universal testing machine. Load–displacement data were recorded at 60 Hz during the experiment. Factors such as yield load, yield displacement, and stiffness were calculated using the load–displacement curve.

The static torsion test was performed similarly to the axial compression test, with a load at a rate of 60°/min until the specimen failed. The torque–angle data were recorded at 60 Hz and factors such as yield torque, yield angle, and stiffness were calculated using the torque–angle curve.

### 2.4. Statistical Analysis

All tests were performed in triplicate, and the results are presented as the mean ± standard deviation (SD) unless otherwise noted. Statistical significance was examined by one-way analysis of variance with Tukey’s post hoc comparison. The analyses were performed using SPSS Statistics software ver.22 (IBM Corp., Armonk, NY, USA). Statistical significance was determined for *p*-values below 0.05. 

## 3. Results 

### 3.1. Morphological Characterization

To confirm the morphology, the various PEEK samples were observed by photography, SEM, and FIB analysis before and after coating. The photographs of various PEEK specimens after coating are shown in Figure 2A. These results showed no significant difference between the PEEK and PEEK-PEG, HA, respectively. However, the surface of PEEK-PDA changed to a dark color, demonstrating a visual difference. Figure 2B shows the morphologies of the surface microstructure of various PEEK specimens before and after coating using a scanning electron microscope. The surface of the PEEK was smooth and there was no processing residue. In contrast, after coating onto PEEK, a change of roughness was demonstrated in the surface due to the various polymers. As shown in Figure 2C, the result of this FIB demonstrates that the surface of various PEEK groups was covered fully by each polymer using the dipping coating technique with PEG, HA, and PDA without requiring any toxic solvent. Moreover, 234 nm (PEEK-PEG), 148.6 nm (PEEK-HA), and 393.7 nm (PEEK-PDA) thick polymer layers were coated onto the PEEK surface without any gaps. 

### 3.2. Surface Modification Analysis 

FT–IR was measured to confirm the formation of a coating layer on the surface of the PEEK specimen for each condition, and the results are shown in Figure 3A. The FT–IR spectrum of the PEEK showed the binding of the carbonyl group (C=O) stretching vibration at 1646 cm^−1^ and the aromatic ring compound at 1590 cm^−1^ and 1480 cm^−1^. By checking the characteristic peaks of 1160 cm^−1^, 948 cm^−1^, 1409 cm^−1^, 1027 cm^−1^, and 1452 cm^−1^, it was confirmed that PEG, HA, and PDA coating layers were formed and detected differently from the control group [24,25,26]. FT–IR analysis determined whether a coating layer was formed according to polymer compositions. Upon comparing the spectral results of PEEK before and after coating, it was observed that there was no significant change in peak detection because the OH, aromatic C=C, and C–N functional groups of PEEK coated with PEG, HA, and PDA were similar [21]. 

To confirm this water ability, Figure 3B shows the contact angle before and after coating using PEG, HA, and PDA onto PEEK. The surface modification is intended to transform the hydrophobic surface of PEEK into hydrophilicity to improve biocompatibility. The contact angle of the uncoated PEEK was 93.64 ± 0.24°, indicating that it had a typical hydrophobic surface. The contact angles of the PEEK-PEG and PEEK-HA surfaces were 64.79 ± 0.53° and 71.64 ± 1.81°, respectively. These results confirmed that the hydrophilicity of the PEEK surface was slightly improved due to the formation of coating layers [21,22]. The contact angle of the PEEK-PDA surface was 32.50 ± 0.05°, which was about three times lower than that of other PEEK groups. Through contact angle analysis, the results of reforming the coated surface into a hydrophilic surface were confirmed. In addition, after coating for each condition, all of the coating specimens were observed to have a lower contact angle than the control group. Therefore, the coating layers of PEG, HA, and PDA conducted in this study formed successfully to improve the hydrophilicity of the surface modification. In particular, in the case of PEEK-PDA with a catecholamine structure, the result of the contact angle was considered to be the lowest among the experimental groups. To confirm the retention of the coating layer of various PEEK groups, Figure 3C shows that the PEKK-PDA had the best retention without change and that PEEK-HA retained the coating layer longer than PEEK-PEG owing to the chemical structure of PEG. This means that PEG has more hydroxyl groups than HA, and therefore it rapidly degraded within DMEM. Otherwise, the PDA result was due to the presence of functional groups, such as carboxyl and amine functional groups, that were found on the surface. This led to the modified surface becoming hydrophilic [25,26].

Subsequently, to analyze the effect of coating, surface roughness was measured to create an average roughness in the range of 0.5 μm to 2.5 μm R_a_. The change value of the various PEEK groups on the surface roughness before and after coating for each condition was measured, and the results are shown in Table 1. The average of the surface roughness R_a_ values of the PEEK before coating was 0.635 μm, and then after coating, the PEEK coated with PEG, HA, and PDA had values of 0.756 μm, 0.744 μm, and 0.803 μm, respectively. Among them, PEEK-PDA had the highest value, which was attributed to the effect of nano-particle formation on the surface according to the PDA coating layer. The R_z_ value, which is a 10-point average illumination, showed a similar tendency [27,28]. So, we assumed that PEEK-PDA would be highly biocompatible among the experimental groups. Next, cell proliferation experiments were performed to confirm this hypothesis.

### 3.3. Biocompatibility Study

As shown in Figure 4, a cell experiment was performed to confirm the activity of various PEEK groups in cell adhesion and proliferation in cultured cells according to each condition of coating layers. Cell viability was confirmed by using the WST-1 assay. These results showed the various PEEK groups with a coating layer compared to the uncoated PEEK. In particular, PEEK-PDA showed the best cell growth rate compared to other groups, demonstrating results that were about twice as high as those of uncoated PEEK. As mentioned in Figure 3, despite the contact angle results of PEEK-PEG being slightly higher than PEEK-HA, the cell viability test revealed that PEEK-HA showed higher cell growth and proliferation than PEEK-PEG due to the retention difference of its coating layer. 

Furthermore, these results of the PEEK-PDA showed the highest cell growth and proliferation because this chemical, which has carboxyl and amine functional groups on the surface, provided an environment with improved cell affinity [29,30]. This allows for cell attachment and the growth of osteoblasts, potentially promoting more rapid bone fusion, which can improve the initial rate of osseointegration of the implant for bone tissue regeneration [31,32,33,34]. 

### 3.4. Mechanical Properties Studies

To confirm the mechanical properties of the stability aspect of the commercial product and the coating product, the results of the compression test according to the coating conditions are shown in Figure 5 and Table 2, according to ASTM F2077. During the test, the distance between the intervertebral discs was set to 6 mm, and the yield displacement was calculated to be 0.12 mm, which is 2% of the distance, according to ASTM F2077. This calculation was used to obtain the yield load. The average yield load, which is the main performance indicator, was observed as follows: 10,072.4 N in PEEK, 10,074.5 N in PEEK-PEG, 1019.1 N in PEEK-HA, and 10,090.8 N in PEEK-PDA. These values were very similar. The values of stiffness for PEEK, PEEK-PEG, PEEK-HA, and PEEK-PDA were 8578.4 N/mm, 8549.6 N/mm, 8561.9 N/mm, and 8487.7 N/mm, respectively. Comparatively, the load of PEEK-PDA was found to be about 14 N higher than that of the control group, but this was not statistically significant. In the case of the PEEK, the standard deviation of the compression test tended to be the largest compared to other groups. In the case of stiffness, the experimental group generally showed lower values than the PEEK, but there was no statistically significant difference. After the compression test, the fracture pattern of the intervertebral fusion prosthesis PEEK also showed the same pattern. In the compression tests before and after coating, it was confirmed that there was no significant difference in yield load, stiffness, or fracture patterns. 

As a result of the static torsion test in Figure 6 and Table 3, the yield torque values were as follows: 6.625 N·m for the control PEEK, 6.427 N·m for PEEK-PEG, 6.635 N·m for PEEK-HA, and 6.585 N·m for PEEK-PDA. These values confirm no significant difference between groups in yield torque and stiffness, which are the main performance indicators. 

## 4. Discussion

Biomaterials used as implants play an important role in spinal reconstruction, fracture fixation, and a variety of bone diseases. There are many bone diseases caused by rapid aging, and various causes are continuing their increase, so the demand for implants is expected to increase further. Implants are required for improved functional role and quality. Recently, PEEK implants were able to supplement the stress shielding problem of titanium implants, but the problem of poor initial osseointegration still remained due to insufficient biocompatibility. 

Previously, we reported on surface modification by polydopamine, bio-mineralization, and BMP-2 immobilization [33]. It was confirmed through the BMP-2 release retention test that the surface modification by the polydopamine coating was maintained for a long period of time. Thus, we designed a biocompatible PEEK implant using a very simple and easy production method with PEG, HA, and PDA to improve initial osseointegration. The coated PEEK groups formed successfully on the modified surface of PEEK, which was investigated by photographic, SEM, and FIB analyses (Figure 2). In particular, when compared to the uncoated PEEK and coating groups including PEEK-PEG, PEEK-HA, and PEEK-PDA, it was observed that the roughness of the surface was increased due to the formation of a coating layer. These rough surfaces promoted cell adhesion and migration, and could provide a good environment for improved cell proliferation and initial osseointegration. Additionally, a morphological change in PEEK-PDA was clearly observed as nanoparticles were formed, and this result indicated that hydrophilicity and biocompatibility could be expected [33]. 

The FT-IR spectra of various PEEK groups exhibited peaks for the chemical composition of coating materials. These results, including Figure 2C, displayed successful PEG, HA, and PDA coating, and the effect of surface changes on the hydrophilicity was then tested further. The hydrophilicity of biomaterials plays an important role in their interactions with biological systems (e.g., protein adsorption and cell attachment). Thus, we provided hydrophilicity coating for the PEEK using PEG, HA, and PDA to improve cell attachment and to enhance cell migration. We expected that PDA would be the most suitable to improve hydrophilicity through the surface modification of PEEK. In a previous report, Park et al. demonstrated that surface modification with PDA chemistry resulted in improved hydrophilicity and good cell proliferation [34]. Similarly, our contact angle results indicated that the PEEK-PDA also exhibited dramatically increased hydrophilicity for the PEEK-based cages. A contact angle of materials in the range of 0° to 40° is favored for bone graft implants, and for this reason our PEEK-PDA cages would be suitable as surgical bone implants. 

Consequently, the properties of materials such as medical devices that can provide a highly cell-friendly environment are required [35,36]. The previous reports mentioned above indicated that, similar to our cell biocompatibility results, the PEEK-PDA showed good cell proliferation as compared with other groups due to its enhanced hydrophilicity. Therefore, these results showed that surface hydrophilicity is an important component for cell adhesion and growth on biomaterials for bone tissue regeneration. Despite PEEK-PEG having more hydrophilicity on the surface than PEEK-HA, the result of the cell viability test showed PEEK-PEG to be lower than PEEK-HA due to the retention of the coating layer. This means that owing to the chemical structure, PEG has more hydroxyl groups than HA, so it was rapidly degraded within DMEM. 

In addition, the PDA-coated substrate can react to combine peptides and many biomolecules, including primary amine and thiol groups, through Michael addition or imine formation. The reason for this is the PDA effect, which involves catechol and amine groups; this advantage permits the formation of the polymer layer on any substrate with a low alkaline pH. This property has useful potential in both bone regeneration and combined biomolecule factors. The above in vitro result suggests that our modified PEEK samples show improved biocompatibility and may play a key role in bone tissue engineering [34]. 

Clinically, in order to achieve successful fusion, the PEEK cage requires the maintenance of its mechanical integrity under a physiological environment. In order to measure the mechanical changes in various PEEK groups in this study, we aimed to compare and analyze the mechanical properties of intervertebral fusion prostheses according to the coating materials of PEEK. We produced specimens by coating each material on commercial products, and static and torsion tests were performed. The value of the range difference investigated did not correlate with PEEK groups in these specimens; it was confirmed that PEG, HA, and PDA coatings only improved biocompatibility through surface modification, without changing the mechanical properties. This is considered to be a method for maintaining high mechanical properties, which is the advantage of PEEK [37,38]. In particular, it is used in intervertebral fusion prostheses, where mechanical properties are important as a support for treating spinal diseases. Also, these results indicated that although there are many previous studies involving Yang et al. [20] and Kwon et al. [21] on PDA coating, our developed PEEK-PDA attempted to differentiate itself by applying a PDA coating to commercial products and verifying the performance of the product by confirming the suitable mechanical properties following ASTM F 2077 for licensing the product. 

Compared to both Yang et al. and Kwon et al. studies, Kwon et al. purposed to add multifunctional properties by dopamine-mediated immobilization of collagen or insulin on the PEEK surface [21]. Yang et al. aimed to accelerate the osseointegration by providing an antibacterial surface onto PEEK [22], but we focused on improving hydrophilicity for increasing the initial cell growth using direct PDA coating, which can provide a rich catechol group and amine group on commercial products of the PEEK cage. Furthermore, we considered that the initial osseointegration was the most important successful fusion within 7 days, so we observed biocompatibility and maintenance of the PDA coating layer for 7 days. Unlike Yang and Kwon, our PEEK-PDA was based on a mechanical analysis that the coating layer did not affect the change. These results are very significant because if mechanical properties are not maintained, they can not be applied to clinical treatment and will not meet the product’s licensing standards. 

In this study, we tried to improve the lack of biocompatibility of PEEK cage by using polymer coating. Through physicochemical tests, PEEK-PDA with dramatically improved surface hydrophilicity among them (PEG, HA, and PDA) showed the highest biocompatibility in an in vitro test. In addition, it was confirmed that the PDA coating layer did not significantly affect the mechanical properties of the product. Thus, the results suggest that PEEK-PDA may have the potential for clinical application in fusion surgery for spinal diseases, as it may improve the rate of osseointegration. Although we have confirmed the effect of PDA through previous research, there is still a limitation in that we have not been able to prove the effect through in vivo tests. So, further studies may be required to confirm the effect through an in vivo test with binding drugs or factors that promote bone formation, such as BMP-2. Also, we plan to develop a mechanical test method to permit this phenomenon and allow for in vivo tests. 

## 5. Conclusions

In this study, surface modification was performed on PEEK by using PEG, HA, and PDA coating techniques to obtain bioactive functions such as hydrophilicity. This modified PEEK-PDA exhibited good roughness and hydrophilic properties compared to the other groups. An in vitro test displayed greatly improved cell adhesion and proliferation on PEEK-PDA about twice as high compared to uncoated PEEK. In addition, mechanical experiments confirmed that there was no significant difference between groups in yield load and stiffness, which are the main performance indicators. The results suggest that our method serves to maintain mechanical properties similar to human bone, which is the advantage of PEEK. Thus, we believe that PEEK-PDA can exhibit effective biocompatibility and suitable mechanical properties in medical devices for orthopedic implants, and that these results would have useful applications in bioengineering.

## Figures and Tables

**Figure 1 bioengineering-11-00343-f001:**
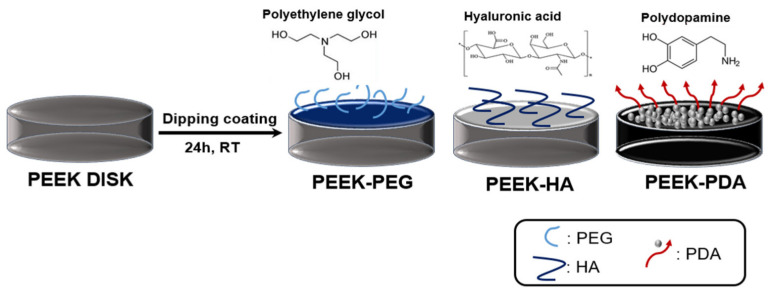
Schematic illustration of the fabrication of the bioactive polyetheretherketone (PEEK) cages for commercial products. The various PEEK cages (PEEK-PEG, PEEK-HA, and PEEK-PDA) were prepared through the dipping coating method, using polyethylene glycol/hyaluronic acid/polydopamine for surface modification steps.

**Figure 2 bioengineering-11-00343-f002:**
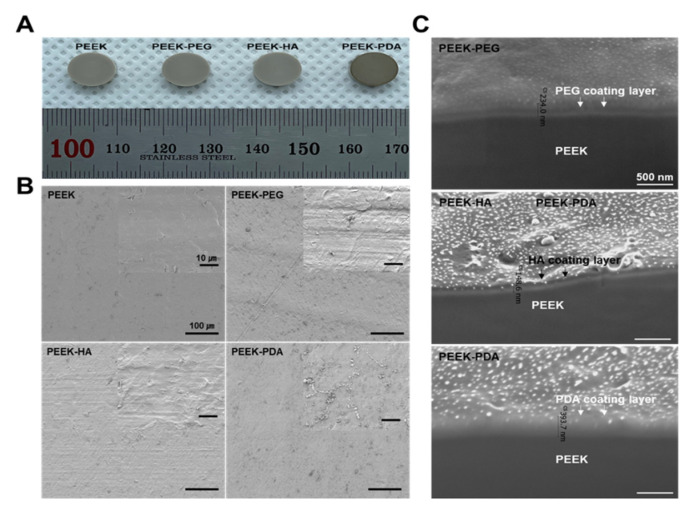
(**A**) Test specimen images after coating, (**B**) scanning electron microscope images, and (**C**) focused ion beam technique images of PEEK, PEEK–PEG, PEEK–HA, and PEEK–PDA.

**Figure 3 bioengineering-11-00343-f003:**
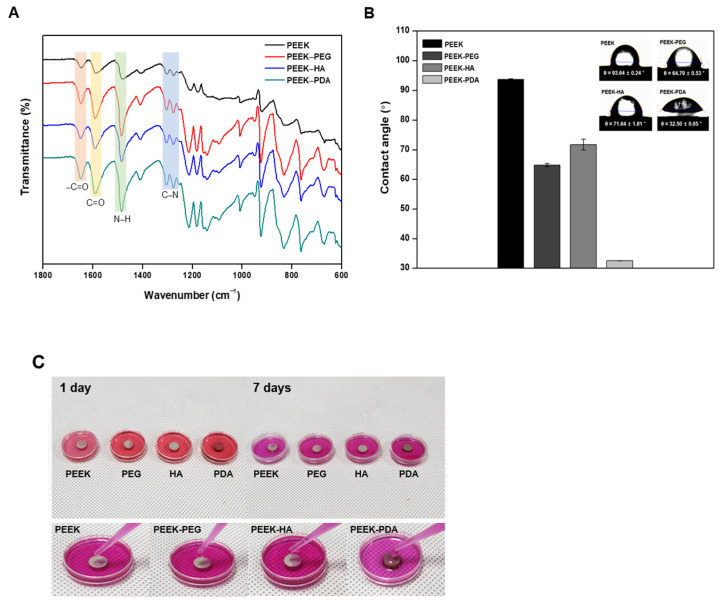
(**A**) The results of Fourier transform infrared spectroscopy, (**B**) the value of water contact angles, and (**C**) the retention results of the coating layers of PEEK, PEEK–PEG, PEEK–HA, and PEEK–PDA for 7 days.

**Figure 4 bioengineering-11-00343-f004:**
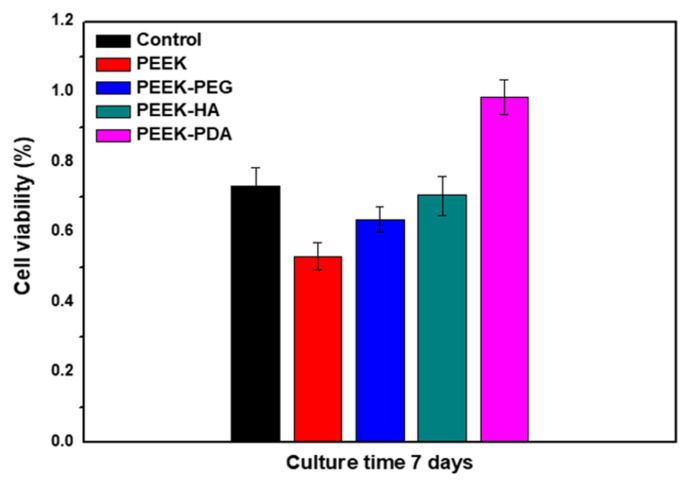
Viability of osteoblast cell growth on the various PEEK formulations for 7 days.

**Figure 5 bioengineering-11-00343-f005:**
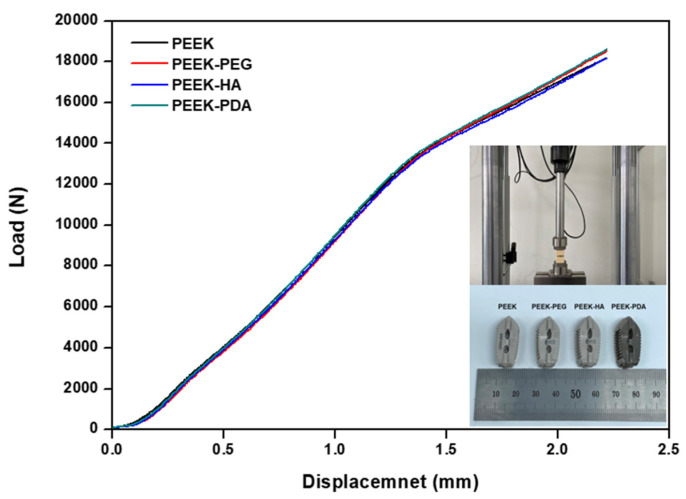
Load–displacement curves of the static compression test according to coating.

**Figure 6 bioengineering-11-00343-f006:**
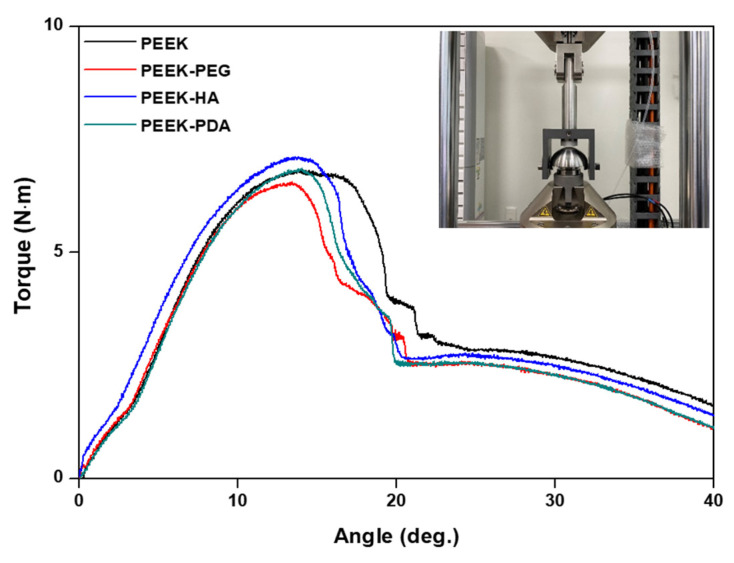
Torque–angle curves of the static torsion test according to coating.

**Table 1 bioengineering-11-00343-t001:** Surface roughness value of PEEK, PEEK-PEG, PEEK-HA, and PEEK-PDA.

Group	R_a_/μm	R_q_/μm	R_z_/μm
PEEK	0.635 ± 1.52	0.771 ± 1.65	4.261 ± 1.77
PEEK-PEG	0.756 ± 1.68	0.924 ± 1.76	4.416 ± 1.87
PEEK-HA	0.744 ± 1.71	0.871 ± 1.69	4.056 ± 1.74
PEEK-PDA	0.803 ± 1.83	0.963 ± 1.72	5.284 ± 1.82

**Table 2 bioengineering-11-00343-t002:** Yield load and stiffness values of the static compression test.

Group	Yield Load (N)		Stiffness (N/mm)	
	Average	Std. Dev.	Average	Std. Dev.
PEEK	10,072.430	179.276	8578.433	196.566
PEEK-PEG	10,074.479	70.874	8549.567	125.066
PEEK-HA	10,019.144	103.655	8561.833	199.118
PEEK-PDA	10,090.790	165.498	8487.683	207.593

**Table 3 bioengineering-11-00343-t003:** Yield torque and stiffness values of the static torsion test.

Group	Yield Torque (N·m)		Stiffness (N·m/deg.)	
	Average	Std. Dev.	Average	Std. Dev.
PEEK	6.525	0.181	0.845	0.045
PEEK-PEG	6.527	0.176	0.803	0.048
PEEK-HA	6.737	0.187	0.804	0.041
PEEK-PDA	6.685	0.184	0.839	0.043

## Data Availability

The data presented in this study are available on request from the corresponding author.

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
