# Peer review of "Surface Modification of Polyetheretherketone (PEEK) Intervertebral Fusion Implant Using Polydopamine Coating for Improved Bioactivity"

_bioengineering, 2024, doi:10.3390/bioengineering11040343_

Round 1

Reviewer 1 Report

Comments and Suggestions for Authors

In the manuscript titled "Surface modification of PEEK (polyetheretherketone) intervertebral fusion implant using polydopamine coating for improved bioactivity" by Suzy Park and Tae Gon Jung, the authors endeavored to enhance the bioactivity of PEEK by modifying its surface through coating with PEG, HA, and PDA. Subsequently, they conducted a comprehensive characterization including physio-chemical, mechanical, and in vitro biocompatibility analyses on both PEEK and PEEK-surface modified samples. The potential application of PDA-coated PEEK in fusion surgery for spinal-related diseases is highlighted. However, the manuscript's overall quality, from abstract to conclusion, is deemed subpar. Therefore, I strongly recommend thorough editing of the manuscript to improve its clarity and presentation, coupled with comprehensive English editing to enhance reader comprehension and elevate the manuscript's quality.

Major comments

Results and Discussion Section:

-        The section contains abundant content but lacks a proper presentation of results and discussion. It requires a rewrite for improved clarity and coherence. Additionally, non-relevant parts should be removed, and repetition of methods should be avoided. For instance:

Cell activity study, section 3.2… The following sentences (lines 311 to 314) are not related to the result or discussion of this section “Biocompatible materials used to develop intervertebral fusion prostheses require high bioactivity to quickly recover and promote the initial rate of osseointegration after surgical treatment into the body. Consequently, the properties of materials such as medical devices that can provide a highly cell-friendly environment are required”….I suggest the authors remove these sentences.

Additionally, the authors repeated the methods parts followed by the above sentence “The PEEK, PEEK-PEG, PEEK-HA, and PEEK-PDA were cultured until 7 days. The cell viability was confirmed by using the WST-1 assay”….I suggest the authors avoid the methods repetition in the result part.

Further, the authors presented that “According to the results of surface modification to hydrophilic the cell growth rate was high in the order of PEG, HA, and PDA coating groups as stated earlier result of the contact angle measurement”. But, in the contact angle measurement, the author showed (Figure 3B) order was PEEK-HA (71.64), PEEK-PEG (64.79), and PEEK-PDA (32.50). If the authors say that cell viability on the PEEK-modified samples was based on hydrophilicity, why did the PEG-coated sample show less viability than the HA-coated samples? Because PEG-coated sample shows better hydrophilicity than HA-coated samples.

In another example (surface roughness part), the authors stated that “the surface average roughness of the PEEK specimen coated with PEG, HA, and PDA was 0.756 µm, 0.744 µm, and 0.803 µm, respectively”. However, in Table 1, the author’s mentioned values were controversial for the above-mentioned group. Please clarify which one is correct.

Minor Comments

-        In the abstract, part of the following sentence was not able to understand what the authors are trying to say, make the sentence very clear. “ In particularly, PEEK-PDA was significantly improved to roughness and water-ability compared to another groups”. 

-        In the Abstract, the authors stated the following sentence “These results confirmed that the large number of carboxyl and amine functional groups derived from the PDA coating layer were consistently retained, forming a hydrophilic surface layer”. It seems there are no clear results presented to prove the above statement.

-         In Figure I, in the box two markings were shown for PDA group, it would be better to show one marking.

-        I suggest the authors present a subsection of the materials and method section very short and neat. For example, give subtitles in such a way.

2.1 Preparation of PEEK

2.2 Preparation of PEEK-modified surface

2.3 Characterizations

      2.31 Phyco-chemical characterization

      2.3.2 Mechanical Characterization

      2.3.3 Other characterizations

2.4 Cell culture and biocompatibility/cytotoxicity, etc (I suggest the authors remove the word “cell activity” from the manuscript. It has a wide meaning).

- Additionally, presents the methods step by step and in a simple language. For example, Section 2.4 is not able to follow it. Make it simple to understand.

-        There are abbreviation and expansion-related errors in the manuscript. Please check it line by line and correct it.

Comments on the Quality of English Language

Moderate editing of English language required

Author Response

We are grateful for the reviewer’s meaningful comment.

Reviewer 2 Report

Comments and Suggestions for Authors

The manuscript titled “Surface modification of PEEK (polyetheretherketone) intervertebral fusion implant using polydopamine coating for improved bioactivity” aimed at modifying the surface of PEEK to become a more hydrophilic surface by coating three different types of polymer: PEG, HA, and PDA.

However, both the methodology used for coating and the selection of modifying agents (PEG, HA, and PDA) do not present a substantial degree of novelty for possible publication in Bioengineering. Actually, there are plenty of previous literature in this domain available, which utilized similar coating approaches to coat HA or HA, or PDA for increasing the compatibility and hydrophilicity of PEEK  (I listed a couple of them below).

[1] Yang, Xin, et al. "A dual-functional PEEK implant coating for anti-bacterial and accelerated osseointegration." Colloids and Surfaces B: Biointerfaces 224 (2023): 113196.

[2] Kwon, Giwan, et al. "Enhanced tissue compatibility of polyetheretherketone disks by dopamine-mediated protein immobilization." Macromolecular Research 26.2 (2018): 128-138.

Given the lack of novelty in this study, the reviewer recommends rejecting the manuscript.

Comments on the Quality of English Language

The English quaility needs moderate improvements.

Author Response

(The authors gave the same response as above.)

Reviewer 3 Report

Comments and Suggestions for Authors

The current research article is representing the surface modifications of PEEK polymer with different types of polymer for intervertebral fusion implant application. Although, the idea is not very novel, but it seems interesting for researchers working in the hard tissue replacements.  However, this research needs a major revision before being accepted in the Bioengineering journal as following:

In the abstract

-          Expand WST-1 in full name

-          Include some numerical results

In the Introduction

-          Move Figure 1 to the methodology section (2.2).

Materials and methods

-          Cite Figure 1 in the text

-          Add more details about the used PEEK such as molecular weight, company, city and town.

In line 144 '' In addition, to confirm the coating layers of PEG, HA, and PDA on the PEEK surface, we were also characterized by the focused ion beam technique (FIB, ThermoFisher, Helios 5 UC, USA).'' , ''we'' should be changed to they.

Results and discussion

-          Section 3.1 needs more discussion.

-          Subheadings are missing for most of characterizations.

-          Water uptake was mentioned in line 266, but this was not mentioned in the methodology section, maybe the authors mean the contact angle.

-          The discussion of the contact angle should be enriched/supported with more references and compared to the previous literatures.

-          Definition of bioactivity is different from biocompatibility and it's much related to ability of implant to uptake Ca and P ions from physiological fluids and form apatite like layer on their surface. Therefore, I biocompatibility term is more suitable for the cell viability made in this research article. The author should replace bioactivity term in the whole manuscript with biocompatibility term.  

-          Figure 4, the cell viability results should be compared with normal cells (control).

-          Table 3 is not cited in the text.

Conclusion

Cell adhesion and proliferations experiments were mentioned in the conclusion but they were not mentioned in any other sections of the whole manuscript.

Author Response

(The authors gave the same response as above.)

Round 2

Reviewer 1 Report

Comments and Suggestions for Authors

In the manuscript titled "Surface modification of PEEK (polyetheretherketone) intervertebral fusion implant using polydopamine coating for improved bioactivity" by Suzy Park and Tae Gon Jung, I appreciate the authors' efforts to address the comments. I am satisfied with authors responses. However, I suggest the authors address the following comments for further consideration.

Comments

-         Please split the following paragraph and merge the first sentence (Thus, there is a critical need for the fatigue testing of degradable materials to evaluate the influence and effects of long-term fatigue) with the previous paragraph. Move the second sentence to the take-home message paragraph (see my comment below).

“Thus, there is a critical need for the fatigue testing of degradable materials to evaluate the influence and effects of long-term fatigue. Future work is planned to develop a mechanical test method to permit this phenomenon and allow for in vivo tests.”

-        The authors should synthesize the main findings of the study and provide insights into their  significance at the end of the discussion (i.e., before the conclusion paragraph). I suggest adding a separate paragraph at the end of the discussion that summarizes key findings, limitations, future directions, and concludes the discussion.

Author Response

(The authors gave the same response as above.)

Reviewer 3 Report

Comments and Suggestions for Authors

The authors have covered most of the comments and the article could be accepted for publication

Author Response

(The authors gave the same response as above.)
